# CT texture analysis of tonsil cancer: Discrimination from normal palatine tonsils

**Tae-Yoon Kim**[1], **Ji Young Lee**[2]�*, **Young-Jun Lee**[2]�*, **Dong Woo Park**[1], **Kyung Tae**[3], **Yun Young Choi**[4]

**1** Department of Radiology, Hanyang University Guri Hospital, Guri, Republic of Korea, **2** Department of Radiology, Hanyang University Hospital, Seoul, Republic of Korea, **3** Department of Otolaryngology-Head and Neck Surgery, Hanyang University Hospital, Seoul, Republic of Korea, **4** Department of Nuclear Medicine, Hanyang University Hospital, Seoul, Republic of Korea

he These authors contributed equally to this work.

* jyjy133@naver.com (JYL); yjleeee@hanyang.ac.kr (YJL)

**Data Availability Statement:** Data cannot be shared publicly because of data contain potentially identifying or sensitive patient information. Data are available from the Hanyang University Hospital and Hanyang University Guri Hospital Institutional

## Abstract

The purposes of the study were to determine whether there are differences in texture analysis parameters between tonsil cancers and normal tonsils, and to correlate texture analysis with $^{18}$F-FDG PET/CT to investigate the relationship between texture analysis and metabolic parameters. Sixty-four patients with squamous cell carcinoma of the palatine tonsil were included. A ROI was drawn, including all slices, to involve the entire tumor. The contralateral normal tonsil was used for comparison with the tumors. Texture analysis parameters, mean, standard deviation (SD), entropy, mean positive pixels, skewness, and kurtosis were obtained using commercially available software. Parameters were compared between the tumor and the normal palatine tonsils. Comparisons were also performed among early tonsil cancer, advanced tonsil cancer, and normal tonsils. An ROC curve analysis was performed to assess discrimination of tumor from normal tonsils. Correlation between texture analysis and $^{18}$F-FDG PET/CT was performed. Compared to normal tonsils, the tumors showed a significantly lower mean, higher SD, higher entropy, lower skewness, and higher kurtosis on most filters (p<0.001). On comparisons among normal tonsils, early cancers, and advanced tonsil cancers, SD and entropy showed significantly higher values on all filters (p<0.001) between early cancers and normal tonsils. The AUC from the ROC analysis was 0.91, obtained from the entropy. A mild correlation was shown between texture parameters and metabolic parameters. The texture analysis parameters, especially entropy, showed significant differences in contrast-enhanced CT results between tumor and normal tonsils, and between early tonsil cancers and normal tonsils. Texture analysis can be useful as an adjunctive tool for the diagnosis of tonsil cancers.

## Introduction

Palatine tonsils are lymphoid tissues that comprise the anterior tonsillar pillar, tonsillar fossa, and posterior tonsillar pillar [1, 2]. Squamous cell carcinoma accounts for about 90% of tonsil

Data Access / Ethics Committee (contact via irb@hyumc.com and hyirbguri@gmail.com) for researchers who meet the criteria for access to confidential data.

**Funding:** The author(s) received no specific funding for this work.

**Competing interests:** The authors have declared that no competing interests exist.

**Abbreviations:** ADC, Apparent diffusion coefficient; CT, Computed tomography; FDG, Fluorodeoxyglucose; MRI, Magnetic resonance imaging; PET/CT, Positron-emission tomography-computed tomography; ROC, Receiver operating characteristic; ROI, Region of interest; SD, standard deviation.

cancer [3]. Patients with advanced tonsil cancer usually present with large oropharyngeal masses and cervical nodal metastasis. However, early tonsil cancer sometimes can be difficult to identify on contrast-enhanced CT because it can have the same appearance as normal lymphoid tissue [4]. Accordingly, the detection of tonsil cancer can be difficult on conventional CT. Therefore, preoperative additional imaging can be needed for the detection of tonsil cancer.

[18]F-FDG PET/CT has been widely used and reported to be valuable in the detection of tonsil cancer, showing higher sensitivity in detecting primary tumors than CT and MR imaging [5]. However, it has the limitation of radiation exposure and false-negative readings. Diffusion weighted image (DWI) has been used to differentiate normal tonsils from tumors [2, 4]. Bathia et al. [4] reported that tonsil cancer shows higher mean apparent diffusion coefficient (ADC) than normal tonsils. Histogram analysis of DWI also showed that the standard deviation of the overall curve is a useful parameter for the detection of occult palatine tonsil cancer [2]. However, the clinical value of histogram analysis of ADC maps remains challenging in daily clinical practice.

Recently, texture analysis has been introduced and applied in lung [6, 7], esophageal [8], colorectal [9], and head and neck cancers [10–12]. It quantifies the tumor heterogeneity using mathematical calculations of spatial patterns or arrangement of pixel intensities [13]. In head and neck cancers, it is usually used for treatment assessment [11, 12] and correlates with human papillomavirus (HPV) status [10, 14]. Until now, it has not been used for the discrimination of tonsil cancer from normal tonsils. Therefore, the purpose of the study was to determine whether there are differences in texture analysis parameters between tonsil cancers and normal tonsils. Additionally, this study aimed to correlate texture analysis with [18]F-FDG PET/CT to investigate the relationship between texture analysis and metabolic parameters and to enhance the understanding of texture analysis results.

## Materials and methods

The Hanyang University Hospital institutional review board and Hanyang University Guri Hospital institutional review board approved the study, and informed consent was waived in accordance with the requirements of a retrospective study. This study included 64 consecutive patients with histopathologically confirmed unilateral tonsillar cancer (38 men, 26 women; age range, 48–88 years; mean age, 59.13 years). Diagnosis was made with tonsillectomy in 44 cases and with biopsy in 20 cases. Patients underwent pretreatment contrast-enhanced neck CT (CECT) between March 2005 and July 2019 in a tertiary care hospital (Hanyang University Hospital) and a secondary care hospital (Hanyang University Guri Hospital). Tumor, node, and metastasis were staged according to the 8th edition of the AJCC staging system. All patients underwent both a CECT and a [18]F-FDG PET/CT. And a gap between CECT and a [18]F-FDG PET/CT was less than 2 weeks.

### CT imaging and texture analysis

CT imaging of 41 patients in Hanyang University Hospital was performed using a 100-s delay, 120 kVp, 200 mAs, 2-mm slice thickness reconstruction (Brilliance 64, Philips Healthcare, Best, The Netherlands; SOMATOM Definition Flash, Siemens Healthcare, Erlangen, Germany). CT imaging of 23 patients at Hanyang University Guri Hospital was performed using the same protocols with 64- or 128-channel scanner systems (SOMATOM Definition DS and SOMATOM Definition Edge, Siemens Healthcare, Erlangen, Germany).

Two neuroradiologists with 7-year and 3-year experiences in the head and neck area performed texture analysis using commercially available software (TexRAD, Cambridge, UK). The freehand region of interest (ROI) was drawn along the outer border of the primary tumor and normal contralateral palatine tonsil on multiple consecutive axial slices to perform volumetric CT texture measurements. Slices with dental artifacts were excluded from the measurements. TexRAD software uses filtration-histogram methods [15], in which image filtering highlights image and object features of a particular size according to the type of filter used. In the spatial scaling factor (SSF, 0, 2, 3, 4, 5, 6), SSF 0 and 2 are defined as *fine*, SSF 3 and 4 as *medium*, and SSF 5 and 6 as *coarse*. Fine filters show enhancement of tissue parenchymal features, whereas medium and coarse filters show enhanced vascular features [16]. The texture analysis-derived parameters were obtained: mean, standard deviation (SD), entropy (texture irregularities and tumor heterogeneity), mean of positive pixels (MPP), kurtosis (peakedness of pixels), and skewness (asymmetry of pixel distribution).

## [18]F-FDG PET/CT image acquisition

All images were obtained with a PET/CT system (GE Discovery, GE healthcare, Waukesha, USA and Siemens, Germany). Patients fasted for at least 6 h before [18]F-FDG PET/CT. Blood glucose level was measured prior to FDG injection and was confirmed to be <180 mg/dL in all patients. Approximately 5.18 MBq/kg [18]F-FDG was intravenously injected 50 min before imaging. First, low-dose CT (120 kVp, tube current modulation) was performed, and a PET scan was obtained from the skull base to the proximal thighs, with an acquisition time of 2.5 min per bed position in three-dimensional mode. PET images were reconstructed with ordered-subset expectation maximization with attenuation correction using vendor-provided software (VUE Point High Definition, GE Healthcare, Milwaukee, WI, USA).

## [18]F-FDG PET/CT image analysis

All PET/CT images were transferred to in-house software (MIM software). The VOI was drawn in the tumor and normal tonsils. Automatically, the metabolic parameters were obtained: standardized uptake values (SUVmax and SUVmean) and total lesion glycolysis (TLG) on the tumor side and normal side. The SUVmaxT/N was additionally obtained after normalization by dividing by the normal side values.

## Statistical analysis

The Kolmogorov–Smirnov test was used to assess the normality of data. Differences in general categorical patient characteristics were analyzed using the χ2 test or Fisher's exact test, as appropriate. The Mann–Whitney U test was performed to assess the difference between normal and palatine tonsils. The differences among the three groups were evaluated using one-way analysis of variance with post hoc analysis or the nonparametric Kruskal-Wallis test with the Mann–Whitney U test. The inter-reader agreement was assessed using the intraclass correlation coefficient (ICC). Receiver operating characteristic (ROC) curve analysis was performed for discrimination of tumors from normal tonsils. Spearman's correlation was obtained to evaluate the linear correlation between texture analysis and metabolic parameters from [18]F-FDG PET/CT. The block diagram of the work is shown in Fig 1.

All statistical calculations were performed using SPSS version 23 (IBM Corporation, Armonk, NY, USA) and MedCalc version 18 (MedCalc, Ostend, Belgium); $p < 0.05$ was considered to be statistically significant.

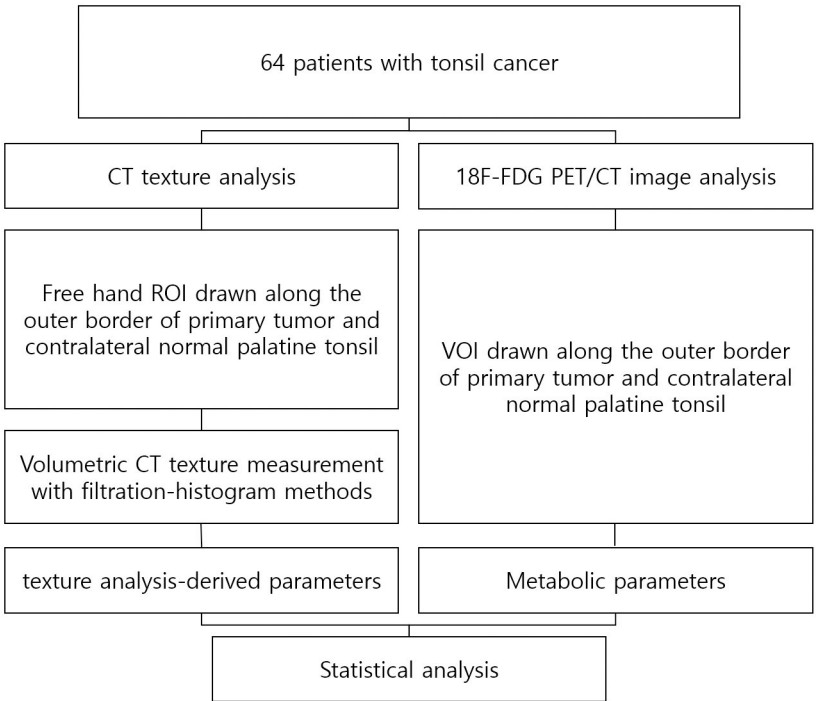

**Fig 1. The block diagram of the work.**

## Results

### General tumor characteristics

There were 14 cancers in the T1 stage, 34 in the T2 stage, 5 in the T3 stage, and 11 in the T4 stage. Forty-eight cancers were early, and 16 were advanced. There were 34 HPV (+) and 18 HPV (-) patients, and 12 with undetermined HPV status. Patient demographic information and staging of the tumors are summarized in Table 1.

### CT texture analysis

The tumor side exhibited a significantly lower mean value with SSF 2–6 than the normal side (p<0.001). The tumor side showed significantly *higher SD and higher entropy* than the normal side at all filter values (p<0.001). The tumor side exhibited significantly *lower skewness* with SSF 0–4 than the normal side (p<0.001, <0.001, <0.001, and = 0.003, respectively). The tumor side showed significantly *higher kurtosis* than the normal side at all filter values (p = 0.001, <0.001, <0.001, <0.001, <0.001, and = 0.012, respectively).

Fig 2 shows the texture analysis-derived features of tonsil cancer, in both early and advanced tumors, compared with contralateral normal tonsils. For the 3 group comparisons, SD, entropy, and kurtosis showed significant differences at all filters (all p<0.001 for SD and entropy; p = 0.005, <0.001, <0.001, <0.001, = 0.001, and = 0.043, respectively, for kurtosis). The mean with SSF 2–6 (p<0.001), and the skewness with SSF 0–4 and 6 (p<0.001, <0.001, <0.001, = 0.012, and = 0.044, respectively), exhibited significant differences. For the comparison between early stage tumors and normal tonsils, early stage tumors showed significantly higher SD, entropy, and kurtosis than normal tonsils at all filter values (all p<0.001 for SD and entropy; p = 0.002, <0.001, <0.001, <0.001, <0.001, and = 0.021 for kurtosis, respectively). Early stage tumors showed significantly lower mean than normal tonsils for SSF 2–6 (p<0.001,

**Table 1. Baseline patient and tumor characteristics.**

|  | Total (n = 64) | Early stage (n = 48) | Advanced stage (n = 16) |
|---|---|---|---|
| Age | 62.32 ± 10.93 | 61.09 ± 11.45 | 65.94 ± 8.44 |
| Sex |  |  |  |
| male | 54 | 39 | 15 |
| female | 10 | 9 | 1 |
| HPV |  |  |  |
| positive | 34 | 25 | 9 |
| negative | 18 | 16 | 2 |
| untested | 12 | 7 | 5 |
| T stage |  |  |  |
| T1 | 14 | 14 | NA |
| T2 | 34 | 34 | NA |
| T3 | 5 | NA | 5 |
| T4 | 11 | NA | 11 |
| N stage |  |  |  |
| N0 | 13 | 11 | 2 |
| N1 | 16 | 15 | 1 |
| N2 | 29 | 21 | 8 |
| N3 | 6 | 1 | 5 |

HPV: Human papilloma virus, NA: not applicable.

<0.001, <0.001, <0.001, and = 0.001, respectively). Early stage tumors showed significantly lower skewness than normal tonsils with SSF 0–4 (p<0.001, <0.001, <0.001, and = 0.021, respectively). In the comparison between the early and advanced tumor groups, only the mean for SSF 3–5 was significantly different (p = 0.032, = 0.022, and = 0.014, respectively). In

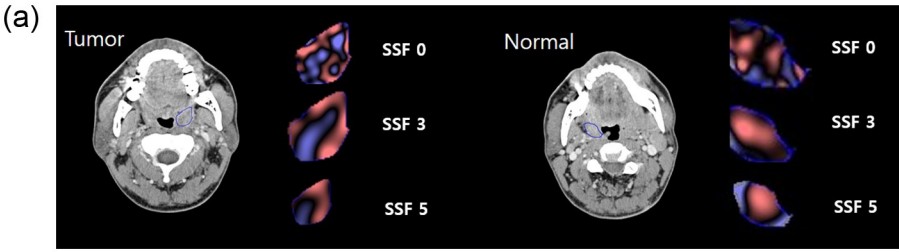

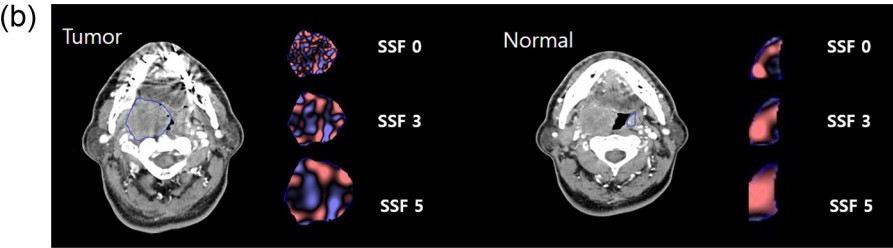

**Fig 2. Imaging appearance and texture analysis features of early stage (a) and advanced stage (b) tonsil cancer.**
ROI for tonsil cancer and contralateral normal tonsils are demonstrated. Representative texture images with SSF 0, 3, 5 and coarse filters are shown. The red color in texture images indicates positive pixel values, and the blue color indicates negative pixel values, which are made after filtration according to filters of different sizes. The tumor side is more heterogeneous than the normal side and tends to have more negative pixel values. This feature is more noticeable with the SSF 0, fine filter.

comparison between the advanced tumor group and the normal tonsil, the mean with SSF 2–6 showed significant differences (p<0.001). SD and entropy exhibited significant differences at all filters (p <0.001, 0.002, <0.001, <0.001, <0.001, <0.001, respectively for SD, and all p<0.001 for entropy). Skewness of SSF 0–3 and 6 showed significant differences (p = 0.006, = 0.002, <0.001, and = 0.043, respectively). Kurtosis of SSF 2–5 showed significant differences (p<0.001, <0.001, <0.001, and = 0.010). All texture analysis parameter values and p values are summarized in Tables 2 and 3. Inter-reader measurement reliability for texture analysis parameters was almost perfect (ICC, 0.81–0.99).

The AUC for entropy with SSF 0 and 4 was the highest with 0.91, differentiating tonsil cancer from normal tonsils (Fig 3). For the entropy with SSF 0, the sensitivity was 79.7% and the specificity was 92.2% at the cutoff value of 4.45. For the entropy with SSF 4, the sensitivity was 90.6% and the specificity was 88.2% at the cutoff value of 5.3. For differentiation of early tonsil cancer from normal tonsils, entropy was the most accurate with an AUC of 0.90, with SSF 0, 3, and 4. For differentiation of advanced tonsil cancer from normal tonsils, entropy was the most accurate with an AUC of 0.94 with SSF 0, 4, and 5.

The correlations between texture analysis and $^{18}$F-FDG PET/CT are shown in S1 Table. There was mild significant correlation between mean and $SUV_{max}$ (r = 0.28, p = 0.04) and between MPP and $SUV_{max}$ (r = 0.27, p = 0.048) at SSF 0. There were also significant correlations between mean and $SUV_{mean}$ (r = 0.3, p = 0.03) and between MPP and $SUV_{mean}$ (r = 0.29, p = 0.03) at SSF 0. Mean showed a negative correlation with TLG at SSF 2, 3, 4, and 5 (r = -0.34, -0.35, -0.34, and -0.31, respectively; p = 0.01). After normalization with the contralateral normal tonsil, entropy showed a linear correlation with $SUV_{max}T/N$ at SSF 2 (r = 0.27, p = 0.04). In addition, entropy showed a mild correlation with $SUV_{mean}T/N$ at SSF 0, 2, 3, and 4 (r = 0.30, 0.36, 0.33, and 0.30; p = 0.03, 0.01, 0.02, 0.03, respectively).

## Discussion

In this study, we found that texture analysis can be useful for differentiating tumors from normal tonsils. Among the texture analysis-derived parameters, entropy was the most helpful in

**Table 2. Comparisons of texture analysis-derived parameters between normal tonsil and tonsil cancer.**

| | Mean | | | SD | | | Entropy | | |
|---|---|---|---|---|---|---|---|---|---|
| SSF | Tumor (n = 64) | Normal (n = 64) | p value | Tumor (n = 64) | Normal (n = 64) | p value | Tumor (n = 64) | Normal (n = 64) | p value |
| 0 | 84.3 | 83.43 | 0.672 | 25.38 | 17.83 | <0.001* | 4.59 | 4.17 | <0.001* |
| 2 | 8.62 | 19.97 | <0.001* | 61.39 | 50.28 | <0.001* | 5.5 | 5.17 | <0.001* |
| 3 | 14.35 | 31.26 | <0.001* | 60.46 | 49.61 | <0.001* | 5.48 | 5.17 | <0.001* |
| 4 | 18.15 | 35.95 | <0.001* | 58.86 | 46.59 | <0.001* | 5.45 | 5.1 | <0.001* |
| 5 | 21.45 | 36.22 | <0.001* | 58.42 | 42.94 | <0.001* | 5.44 | 5.02 | <0.001* |
| 6 | 23.05 | 39.02 | <0.001* | 57.29 | 40.92 | <0.001* | 5.44 | 4.99 | <0.001* |
| | **MPP** | | | **Skewness** | | | **Kurtosis** | | |
| SSF | Tumor (n = 64) | Normal (n = 64) | p value | Tumor (n = 64) | Normal (n = 64) | p value | Tumor (n = 64) | Normal (n = 64) | p value |
| 0 | 85.79 | 83.72 | 0.306 | -0.88 | -0.31 | <0.001* | 2.62 | 1.08 | 0.001* |
| 2 | 48.57 | 47.53 | *0.365 | -0.29 | 0.2 | <0.001* | 1.27 | 0.38 | <0.001* |
| 3 | 51.06 | 53.22 | *0.750 | -0.33 | 0.3 | <0.001* | 0.98 | -0.2 | <0.001* |
| 4 | 53 | 55.29 | *0.804 | -0.31 | -0.14 | 0.003* | 0.79 | 0.1 | <0.001* |
| 5 | 53.32 | 56.07 | *0.990 | -0.31 | -0.29 | 0.801 | 0.4 | -0.11 | <0.001* |
| 6 | 53.46 | 57.92 | *0.712 | -0.31 | -0.41 | 0.141 | 0.16 | -0.14 | 0.012* |

* Statistically significant.

SD standard deviation, MPP mean positive pixel, SSF spatial scaling factor.

**Table 3. Comparisons of texture analysis-derived parameters among normal tonsil, early tonsil cancer, and advanced tonsil cancer.**

| | SSF | Mean | SD | Entropy | MPP | Skewness | Kurtosis |
|---|---|---|---|---|---|---|---|
| **p value** | 0 | 0.635 | <0.001* | <0.001* | 0.444 | <0.001* | 0.005* |
| | 2 | <0.001* | <0.001* | <0.001* | 0.569 | <0.001* | <0.001* |
| | 3 | <0.001* | <0.001* | <0.001* | 0.763 | <0.001* | <0.001* |
| | 4 | <0.001* | <0.001* | <0.001* | 0.771 | 0.012* | <0.001* |
| | 5 | <0.001* | <0.001* | <0.001* | 0.809 | 0.861 | 0.001* |
| | 6 | <0.001* | <0.001* | <0.001* | 0.833 | 0.044* | 0.043* |
| **Early vs normal** | 0 | 0.997 | <0.001* | <0.001* | 0.786 | <0.001* | 0.002* |
| | 2 | <0.001* | <0.001* | <0.001* | 0.312 | <0.001* | <0.001* |
| | 3 | <0.001* | <0.001* | <0.001* | 0.930 | <0.001* | <0.001* |
| | 4 | <0.001* | <0.001* | <0.001* | 0.953 | 0.021* | <0.001* |
| | 5 | <0.001* | <0.001* | <0.001* | 0.899 | 0.909 | <0.001* |
| | 6 | 0.001* | <0.001* | <0.001* | 0.661 | 0.749 | 0.021* |
| **Early vs advanced** | 0 | 0.669 | 0.556 | 0.927 | 0.727 | 0.871 | 0.745 |
| | 2 | 0.072 | 0.988 | 1.000 | 0.515 | 0.420 | 0.471 |
| | 3 | 0.032* | 0.871 | 0.935 | 0.438 | 0.281 | 0.871 |
| | 4 | 0.022* | 0.614 | 0.442 | 0.369 | 0.973 | 0.858 |
| | 5 | 0.014* | 0.525 | 0.372 | 0.321 | 0.974 | 0.914 |
| | 6 | 0.20 | 0.520 | 0.567 | 0.485 | 0.214 | 0.975 |
| **Advanced vs normal** | 0 | 0.622 | <0.001* | <0.001* | 0.432 | 0.006* | 0.43 |
| | 2 | <0.001* | 0.002* | <0.001* | 0.829 | 0.002* | <0.001* |
| | 3 | <0.001* | <0.001* | <0.001* | 0.532 | <0.001* | <0.001* |
| | 4 | <0.001* | <0.001* | <0.001* | 0.613 | 0.099 | <0.001* |
| | 5 | <0.001* | <0.001* | <0.001* | 0.819 | 0.878 | 0.010* |
| | 6 | <0.001* | <0.001* | <0.001* | 0.971 | 0.043* | 0.111 |

* Statistically significant.

SD standard deviation, MPP mean positive pixel, SSF spatial scaling factor.

distinguishing early tonsil cancer from normal tonsils. In cases of early tonsil cancer, which could be difficult to detect with visual analysis on contrast-enhanced CT, the entropy derived from the texture analysis could be useful for the diagnosis of tonsil cancer. There was a mild correlation between texture analysis parameters and FDG uptake on [18]F-FDG PET/CT, which suggests that the texture analysis parameters relate to the physiologic state of the tumor.

Tumor heterogeneity is a well-recognized histopathologic feature of malignancy, reflecting high cell density, necrosis, hemorrhage, and myxoid change [6, 17]. High tumoral heterogeneity is associated with adverse biology, aggressive clinical course, and increased resistance to treatment [18]. A recent study on oropharyngeal cancer showed a correlation between texture and HPV status, showing a lower value of entropy and SD in the HPV-positive group [10]. For the correlation between texture and treatment response, texture analysis parameters are known to be associated with local failure in patients with head and neck cancer with chemoradiotherapy. CT texture features correlated with TNM stages in gastric cancer and esophageal cancer, indicating that tumors with higher T or N stages have higher levels of heterogeneity-related features [19, 20]. Furthermore, several texture features can discriminate between high- and low-grade lung cancers [21, 22]. Meyer et al. reported that CT entropy was correlated with hypoxic related pathological parameter, hypoxia-inducible factor-1-alpha expression [23]. This study was concordant with previous studies that showed high entropy in malignant tumors [10].

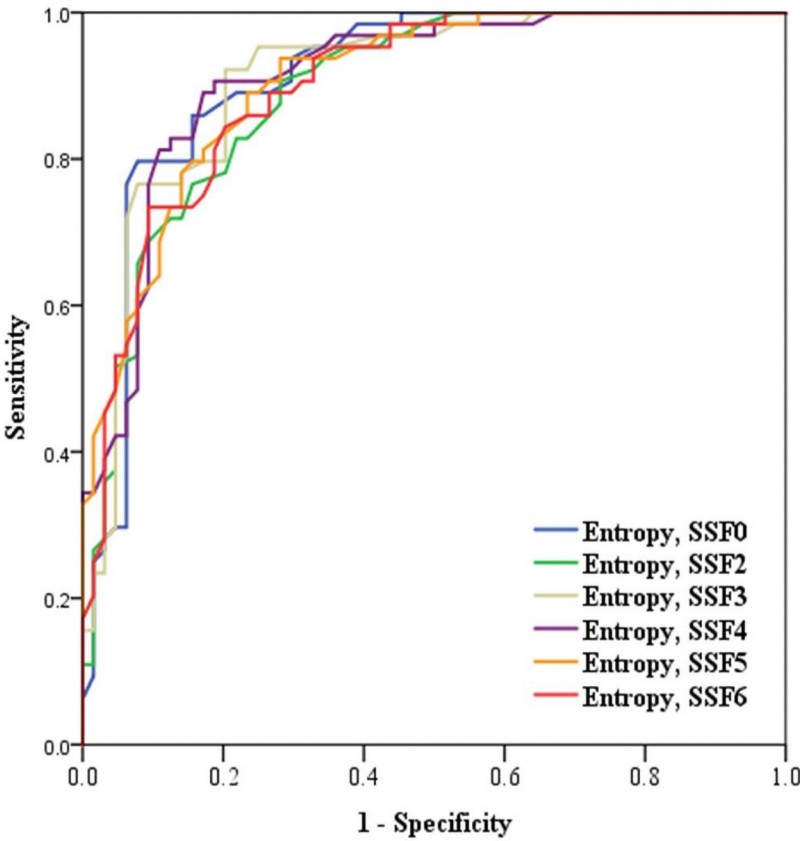

**Fig 3. The ROC curve analysis was performed to differentiate tonsil cancer from normal tonsils.** The AUC for entropy with SSF 0 and 4 was the highest at 0.91.

Texture analysis is a computer-based, image processing technique that allows for the mathematical detection of changes in pixel density, which may be visually imperceptible [6]. It can assess the tumoral heterogeneity by analyzing the distribution and relationship of the pixel gray level in the images. It can maximize the information obtained from routinely acquired diagnostic images in current clinical practice without additional acquisition of images or an invasive procedure. However, the filtration histogram-technique is required owing to the CT photon-noise, which can influence the radiologist's impression of image quality and mask the biologic heterogeneity. By using filters, CT texture analysis reduces the effect of photon noise and enhances the biological heterogeneity [7]. Fine filters usually enhance tissue parenchymal features, and medium to coarse filters enhance vascular features [24].

Tonsil cancer usually demonstrates asymmetric enlargement on contrast-enhanced CT. In cases of early tonsil cancer, the cancer density usually appears similar to that of normal tonsils. Therefore, the visual analysis sometimes fails to depict the mass in the tonsil because there is no difference in enhancement between tonsil cancer and normal palatine tonsils. In these cases, the patients are classified as having a malignancy of unknown origin presenting with cervical nodal metastasis. These patients undergo biopsies of the tonsil, and tongue base to determine the origin of the primary tumor [25]. Accordingly, the hypothesis was that computer-based assessment of texture analysis could be useful in detecting tonsil cancer, especially in the early stage. The study results showed that entropy was the most significant parameter for detecting tonsil cancer, compared with normal palatine tonsils. Entropy refers to the

randomness of pixel intensity. Therefore, in cases of early tonsil cancer, which could be difficult to detect with visual analysis on contrast-enhanced CT, the entropy derived from the texture analysis could be useful for the diagnosis of tonsil cancer.

It can also be difficult to differentiate tonsil cancer from normal lymphoid tissue using magnetic resonance imaging. DWI is an MR technique that has the potential to improve the detection of head and neck cancer [26]. It is well known that malignant tumors have lower ADC values than benign lesions. Multiple studies have provided ADC threshold values, but they have showed a broad spectrum [27–29]. A recent meta-analysis reported that DWI/ADC alone cannot be used as an imaging biomarker of malignancy in head and neck cancer [30].

The correlation analysis to evaluate the relationship between texture analysis parameter and metabolic parameter on [18]F-FDG PET/CT was performed. Generally, FDG uptake by tumors on [18]F-FDG PET/CT shows a correlation with malignancy and reflects the physiologic condition of the tumor [31]. Texture analysis parameters are already known to be related to tumor biology. Texture analysis showed a mild linear correlation with the tumor uptake on [18]F-FDG PEC/CT in this study. This result was concordant with the study by Ganeshan et al [32], which showed a strong correlation between SUVmax or SUVmean and entropy in esophageal cancer.

This study has several limitations. First, the texture analysis was performed in only patients with tonsil cancers. To verify the texture analysis based on tumor detection, further studies with patients with head and neck cancer could be needed with prospective and multicenter studies. Second, a correlation between texture parameters and histopathologic markers, such as angiogenesis or hypoxia, was not performed. To better understand the tumor biology, a future correlation study with histopathologic markers is needed. Finally, despite the scan protocols were same, there might be a potential influence of different CT scanner on the texture analysis.

## Conclusion

In conclusion, entropy was the most useful parameter to distinguish tonsil cancer from normal tonsils. Texture analysis showed a mild linear correlation with tumor uptake on [18]F-FDG PET/CT. Therefore, as a measurement method of tumor heterogeneity, texture analysis relates to the physiologic condition of the tumor. In cases of early tonsil cancer, which could be difficult to detect with visual analysis on contrast-enhanced CT, the entropy derived from the texture analysis could be useful for the diagnosis of tonsil cancer.

## Supporting information

**S1 Table. Results of correlation between texture analysis and [18]F-FDG PET/CT parameters.**
(PDF)

## Author Contributions

**Conceptualization:** Ji Young Lee.

**Data curation:** Tae-Yoon Kim, Young-Jun Lee, Dong Woo Park, Kyung Tae, Yun Young Choi.

**Formal analysis:** Tae-Yoon Kim, Ji Young Lee.

**Investigation:** Dong Woo Park, Yun Young Choi.

**Methodology:** Ji Young Lee.

**Supervision:** Ji Young Lee.

**Writing – original draft:** Tae-Yoon Kim, Ji Young Lee.

**Writing – review & editing:** Ji Young Lee, Young-Jun Lee, Dong Woo Park, Kyung Tae, Yun Young Choi.

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
