## [Decision Letter · Decision Letter 0]

18 Jun 2021

PONE-D-21-07708

CT texture analysis of tonsil cancer: discrimination from normal palatine tonsils

PLOS ONE

Dear Dr. Lee,

Thank you for submitting your manuscript to PLOS ONE. After careful consideration, we feel that it has merit but does not fully meet PLOS ONE’s publication criteria as it currently stands. Therefore, we invite you to submit a revised version of the manuscript that addresses the points raised during the review process.

We look forward to receiving your revised manuscript.

Kind regards,

Thomas Pyka

Academic Editor

PLOS ONE

Journal Requirements:

3. Please ensure that you include a title page within your main document. We do appreciate that you have a title page document uploaded as a separate file, however, as per our author guidelines (http://journals.plos.org/plosone/s/submission-guidelines#loc-title-page) we do require this to be part of the manuscript file itself and not uploaded separately.

4. Thank you for including your ethics statement:  "Our institutional review board approved the study, and informed consent was waived in accordance with the requirements of a retrospective study. ".  

For additional information about PLOS ONE ethical requirements for human subjects research, please refer to http://journals.plos.org/plosone/s/submission-guidelines#loc-human-subjects-research."

5. In your Data Availability statement, you have not specified where the minimal data set underlying the results described in your manuscript can be found. PLOS defines a study's minimal data set as the underlying data used to reach the conclusions drawn in the manuscript and any additional data required to replicate the reported study findings in their entirety. All PLOS journals require that the minimal data set be made fully available. For more information about our data policy, please see http://journals.plos.org/plosone/s/data-availability. "Upon re-submitting your revised manuscript, please upload your study’s minimal underlying data set as either Supporting Information files or to a stable, public repository and include the relevant URLs, DOIs, or accession numbers within your revised cover letter. For a list of acceptable repositories, please see http://journals.plos.org/plosone/s/data-availability#loc-recommended-repositories. Any potentially identifying patient information must be fully anonymized. Important: If there are ethical or legal restrictions to sharing your data publicly, please explain these restrictions in detail. Please see our guidelines for more information on what we consider unacceptable restrictions to publicly sharing data: http://journals.plos.org/plosone/s/data-availability#loc-unacceptable-data-access-restrictions. Note that it is not acceptable for the authors to be the sole named individuals responsible for ensuring data access. We will update your Data Availability statement to reflect the information you provide in your cover letter.

Reviewers' comments:

Reviewer's Responses to Questions

**Comments to the Author**

1. Is the manuscript technically sound, and do the data support the conclusions?

Reviewer #1: Yes

2. Has the statistical analysis been performed appropriately and rigorously? 

Reviewer #1: Yes

3. Have the authors made all data underlying the findings in their manuscript fully available?

Reviewer #1: Yes

4. Is the manuscript presented in an intelligible fashion and written in standard English?

Reviewer #1: Yes

5. Review Comments to the Author

Reviewer #1: comments:

· The statistical analysis in this paper is suitable

· In terms of experimental technique, this paper is novel

· I would like to see some discussion of the findings of the papers with existing values (entropy) in other papers

Minor comments:

· In several places, you've used the term “we”, but it seems unnecessary.

· The author can add Block Diagram of the Work.

If the above changes made in manuscript, I recommend to publish the same.

6. PLOS authors have the option to publish the peer review history of their article (what does this mean?). If published, this will include your full peer review and any attached files.

Reviewer #1: No

---

## [Author Response · Author response to Decision Letter 0]

10 Jul 2021

Point by point responses.

Thank the editor and reviewers for their very thoughtful suggestions and comments that were very helpful for revising the manuscript. Followings are descriptions on the point-by-point corrections made in the revised manuscript and answers to the questions raised by the reviewers. 

Reviewer(s)' Comments to Author:

**

Reviewer #1: comments:

· The statistical analysis in this paper is suitable.

· In terms of experimental technique, this paper is novel

Response: Thanks for the kind review and words. 

· I would like to see some discussion of the findings of the papers with existing values (entropy) in other papers

Response: in response to the comment, one more reference was added as follows: Meyer et al. reported that CT entropy was correlated with hypoxic related pathological parameter, hypoxia-inducible factor-1-alpha expression. 

Minor comments:

· In several places, you've used the term “we”, but it seems unnecessary.

Response: in response to the comment, we corrected most of sentences.

· The author can add Block Diagram of the Work.

Response: We added as table 1 as follows.

If the above changes made in manuscript, I recommend to publish the same

---

## [Editor Report · Decision Letter 1]

26 Jul 2021

CT texture analysis of tonsil cancer: discrimination from normal palatine tonsils

PONE-D-21-07708R1

Dear Dr. Lee,

We’re pleased to inform you that your manuscript has been judged scientifically suitable for publication and will be formally accepted for publication once it meets all outstanding technical requirements.

Kind regards,

Thomas Pyka

Academic Editor

PLOS ONE
---

## [Editor Report · Acceptance letter]

2 Aug 2021

PONE-D-21-07708R1 

CT texture analysis of tonsil cancer: discrimination from normal palatine tonsils 

Dear Dr. Lee:

I'm pleased to inform you that your manuscript has been deemed suitable for publication in PLOS ONE. Congratulations! Your manuscript is now with our production department. 

Kind regards, 

on behalf of

Dr. Thomas Pyka 

Academic Editor

PLOS ONE